# Targeting Wnt Signaling in Endometrial Cancer

**DOI:** 10.3390/cancers13102351

**Published:** 2021-05-13

**Authors:** Iram Fatima, Susmita Barman, Rajani Rai, Kristina W. Thiel, Vishal Chandra

**Affiliations:** 1Department of Biochemistry and Molecular Biology, University of Nebraska Medical Center, Omaha, NE 68198, USA; irambiotech08@gmail.com (I.F.); susmita.barman1987@gmail.com (S.B.); 2Division of Endocrinology, Central Drug Research Institute, Uttar Pradesh, Lucknow 226031, India; 3Department of Biochemistry, CSIR—Central Food Technological Research Institute, Mysore, Karnataka 570020, India; 4Stephenson Cancer Center, University of Oklahoma Health Sciences Center, Oklahoma City, OK 73104, USA; rrai@ouhsc.edu; 5Department of Obstetrics and Gynecology and the Holden Comprehensive Cancer Center, University of Iowa, Iowa City, IA 52242, USA; kristina-thiel@uiowa.edu; 6Division of Gynecologic Oncology, Department of Obstetrics and Gynecology, College of Medicine, University of Oklahoma Health Sciences Center, Oklahoma City, OK 73104, USA

**Keywords:** endometrial cancer, Wnt signaling, β-catenin, mutations, targeted therapy

## Abstract

**Simple Summary:**

Wnt has diverse regulatory roles at multiple cellular levels and numerous targeting points, and aberrant Wnt signaling has crucial roles in carcinogenesis, metastasis, cancer recurrence, and chemotherapy resistance; based on these facts, Wnt represents an appealing therapeutic target for cancer treatment. Although preclinical data supports a role for the Wnt signaling pathway in uterine carcinogenesis, this area remains understudied. In this review, we identify the functions of several oncogenes of the Wnt/β-catenin signaling pathway in tumorigenesis and address the translation approach with potent Wnt inhibitors that have already been established or are being investigated to target key components of the pathway. Further research is likely to expand the potential for both biomarker and cancer drug development. There is a scarcity of treatment choices for advanced and recurrent endometrial cancer; investigating the sophisticated connections of Wnt signaling networks in endometrial cancer could address the unmet need for new therapeutic targets.

**Abstract:**

This review presents new findings on Wnt signaling in endometrial carcinoma and implications for possible future treatments. The Wnt proteins are essential mediators in cell signaling during vertebrate embryo development. Recent biochemical and genetic studies have provided significant insight into Wnt signaling, in particular in cell cycle regulation, inflammation, and cancer. The role of Wnt signaling is well established in gastrointestinal and breast cancers, but its function in gynecologic cancers, especially in endometrial cancers, has not been well elucidated. Development of a subset of endometrial carcinomas has been attributed to activation of the APC/β-catenin signaling pathway (due to β-catenin mutations) and downregulation of Wnt antagonists by epigenetic silencing. The Wnt pathway also appears to be linked to estrogen and progesterone, and new findings implicate it in mTOR and Hedgehog signaling. Therapeutic interference of Wnt signaling remains a significant challenge. Herein, we discuss the Wnt-activating mechanisms in endometrial cancer and review the current advances and challenges in drug discovery.

## 1. Introduction

The human endometrium is a proliferative, angiogenic, and dynamically regenerated normal tissue that is the anatomic prerequisite for pregnancy. Stem/progenitor cells (epithelial, mesenchymal, endothelial) likely contribute to both the rapid endometrial cyclic growth and regeneration of 4–14 mm of mucosa per cycle during a woman’s reproductive lifespan and endometrial regeneration/repair postpartum. During the reproductive life of women, the human endometrium undergoes dynamic cyclical changes with ~400 cycles of regeneration, differentiation, and shedding (menstruation). These cycles are regulated by many signaling pathways but primarily by estrogen and progesterone hormone signals [1]. Estrogen is a classical proliferative signal in the endometrium whereas progesterone counteracts the effects of estrogen to promote cellular differentiation. This dynamic balance between the pro-growth and pro-differentiation effects of estrogen and progesterone, respectively, is essential to prevent abnormal endometrial proliferation. Indeed, several gynecological diseases are associated with abnormal endometrial proliferation, including endometriosis, adenomyosis, endometrial hyperplasia, and endometrial cancer [2,3]. Tumors that develop from the single layer of epithelial cells that line the endometrium and form the endometrial glands are classified as endometrial carcinomas (98% of endometrial tumors) while tumors that develop in the muscle layer (myometrium) or stromal tissue (supporting connective tissue) within the muscle layer are called sarcomas (2%) [4].

Endometrial cancer is the sixth most common gynecologic malignancy of the female reproductive organs and, while rates vary by geographic region, overall, it afflicts approximately 27.8 per 100,000 women worldwide annually [5]. In the US alone, over 65,000 women were diagnosed with endometrial cancer in 2020 with 12,500 deaths. Endometrial cancer is one of only two cancers for which both incidence and mortality are on the rise as compared to 40 years ago [6], and this increase can be attributed, in part, to the obesity epidemic [7]. Endometrial cancer incidence is particularly high in black females in the US with a disparately high mortality compared to other racial and ethnic groups [8].

Genetic alterations as well as various cell-signaling pathways have been implicated in endometrial cancer development and progression. These pathways include Wnt/β-catenin signaling cascades (together with APC/β-catenin signaling), the PI3K/AKT/mTOR signaling pathway, the MAPK/ERK pathway, the ErbB signaling pathway, the VEGF/VEGFR ligand–receptor signaling pathway, and the p53-P16INK4a signaling pathway. Endometrial tumor tissues have been shown to contain mutations in these signaling pathways, which are generally regarded as primary drivers of carcinogenesis [9]. A better understanding of these signaling transduction pathways as well as cross-connections between them is necessary in order to understand the mechanisms of action of existing targeted therapy drugs and to discover new therapeutic applications [10,11].

The Wnt/β-catenin signaling pathway is an extremely conserved pathway that is involved in a variety of cellular processes in the female genital system, including development, cell proliferation, cell survival, adhesion, and motility as well as the regulation of the menstrual cycle. In addition to its crucial role in tissue homeostasis, aberrant Wnt signaling plays a significant role in many diseases from cancer to metabolic disorders [12,13,14,15,16,17]. During the reproductive life in the endometrium, Wnt/β-catenin signaling is regulated by estrogen and progesterone hormones. In the proliferative phase of the menstrual cycle, estrogen stimulates Wnt/β-catenin signaling and enhances the nuclear accumulation of β-catenin. However, later, during the secretary phase, progesterone counterbalances estrogen-induced proliferation by inhibition of Wnt/β-catenin signaling [18,19,20]. Aberrant regulation of the Wnt signaling pathway in the endometrium results in endometrial hyperplasia, which may proceed to endometrial cancer [21,22,23].

Early studies into the role of Wnt signaling in endometrial cancers mainly focused on identifying β-catenin gene mutations. However, mutation of APC (Adenomatous polyposis coli) or CTNNB1 (the gene that encodes β-catenin) is rare in endometrial cancer, indicating that other mechanisms are responsible for the aberrant activation of β-catenin [24,25,26,27]. Moreover, nuclear β-catenin staining is prominent in the early onset of endometrial cancer, endometrial hyperplasia, and well-differentiated endometrioid carcinomas [28,29]. Using the Cancer Genome Atlas (TCGA) database, we observed increased expression of β-catenin in human endometrial cancer patients (Figure 1A,B). Interestingly, these findings were more common in the endometrioid histologic type of endometrial cancers than in nonendometrioid endometrial cancers. Similarly, multiple studies show that inhibition of several negative regulators of the Wnt pathway (by epigenetic silencing), such as members of the soluble Frizzled protein (SFRP) and Dickkopf (DKK) families, may contribute to development and progression of endometrial cancer [30,31,32,33,34,35,36,37,38,39,40].

Unlike for other cancers, the mechanism of Wnt signaling participation in endometrial cancer has not been elucidated and is not limited to the involvement of β-catenin and APC mutations. In this review, we present an overview of the Wnt signaling pathway and its activating mechanisms in endometrial cancer. We will address the common Wnt pathway-associated mutations identified in endometrial cancer and will further review the current therapeutic options targeting Wnt signaling considering both their potential and their limitations.

## 2. Wnt Signaling

The Wnt signaling pathway is evolutionarily highly conserved and is a crucial cascade regulating development and stemness. Wnt signaling is also firmly associated with several cancers. This signaling network can be divided into two modes based on the role of β-catenin: the β-catenin-dependent pathway is called “canonical Wnt/β-catenin signaling” and the β-catenin-independent pathway is called the “noncanonical pathway.” The noncanonical Wnt/β-catenin pathway can be further subdivided into the planar cell polarity (PCP) pathway and the Wnt/Ca2+ pathway [41,42,43] (Figure 2).

In both the canonical and noncanonical pathways, signaling is initiated by binding of Wnt ligands to the extracellular cysteine-rich domain (CRD) at the amino terminus of Frizzled receptors (Fzd) and a number of recently linked coreceptors, including receptor tyrosine kinase-like orphan receptor 1 (ROR1), receptor tyrosine kinase-like orphan receptor 2 (ROR2), and receptor-like tyrosine kinase (Ryk). This ligand–receptor interaction activates canonical WNT/β-catenin and noncanonical WNT/PCP and WNT/Ca2+ signaling pathways. This mechanism is interfered with by several inhibitors and gatekeeper molecules, including the families of SFRPs and DKKs [21,44,45].

The canonical Wnt/β-catenin pathway is activated by binding of Wnt to a transmembrane receptor complex that is formed from the seven-pass transmembrane Fzd and the co-receptor low-density lipoprotein receptor-related protein 5 or 6 (LRP5/6); binding is enhanced by the R-spondin/Lgr interaction. This Wnt–Fzd–LRP6 complex recruits scaffolding protein Dishevelled (Dvl), which leads to the phosphorylation of LRP6 and recruitment of the Axin complex to the receptors. This signaling cascade hinders Axin-mediated β-catenin phosphorylation and subsequently stabilization of β-catenin. Liberated β-catenin then accumulates and translocates to the nucleus where it binds to T cell factor and Lymphoid enhancer-binding factor 1 (TCF/LEF) and promotes the transcription of Wnt target genes, such as *JUN, c-Myc, CCND1*(Cyclin D1), and *AXIN2* among others [46].

As mentioned above, the β-catenin-independent or noncanonical pathway can be further divided into two different branches, the planar cell polarity (PCP) pathway and the Wnt/Ca2+ pathway, both of which are activated by Wnt. The PCP pathway regulates cell motility and polarity through the activation of small GTPases, RhoA, Rac, and the c-Jun N-terminal kinase (JNK). The PCP pathway can be demonstrated by Wnt–Fzd interaction; however, receptor tyrosine kinase-like orphan receptors (RORs) like ROR1, ROR2, and receptor tyrosine kinase (Ryk) can also serve as Wnt receptors to activate β-catenin-independent pathways [47,48,49,50]). The Wnt/Ca2+pathway is activated upon Wnt ligation and leads to an increase in intracellular Ca2+ levels. This signaling comprises the activation of protein kinase C (PKC) and Ca2+/calmodulin-dependent protein kinase II (CAMKII) through the mobilization of DVL and Phospholipase C (PLC) and the subsequent increase in intracellular calcium (Ca2+) release from the endoplasmic reticulum (ER). The Wnt/Ca2+ pathway also signals through protein phosphatase calcineurin (Caln), stimulating the transcription factor nuclear factor of activated T cells (NFAT) [42,43,46].

Wnt/β-catenin pathway activation is a pivotal regulator of epithelial-mesenchymal transition (EMT) [51,52,53]. Wnt signaling is associated with increased expression of Snail, Slug and Twist, which control downregulation of the adherens junction protein epithelial (E)-cadherin and upregulation of mesenchymal specific marker neuronal (N)-cadherin during EMT, promoting cell migration. During early research, EMT was first identified in the setting of normal cell differentiation but is now recognized as a key factor in many different types of cancers, including endometrial cancer [51,54].

## 3. Aberrations in Wnt Signaling in Cancer

The role of aberrant Wnt signaling in cancer was first discovered during tumor induction in a mouse model following proviral integration at the WNT1 locus [55,56]. In recent decades, aberrant activation of Wnt signaling has also been documented in the proliferation, survival, and ability to metastasize in various cancer cell types. Continuous activation of Wnt signaling can be accomplished through either mutational or non-mutational alterations [12].

It is very well reported that aberrant activation of the canonical Wnt/β-catenin signaling pathway is associated with the development of colorectal cancer through mutations in the *APC* and *CTNNB1* (β-catenin) genes that lead to hyperactivation of the pathway. Mutations of β-catenin that result in Wnt signaling dysfunction have frequently been observed in many other cancers, including endometrial cancer, ovarian cancer, medulloblastoma, hepatocellular carcinoma, and Wilms’ tumor, with many studies supporting the idea that elevated β-catenin levels are associated with poor prognosis [12,57].

Furthermore, β-catenin is not the only important component of the Wnt signaling network. Several studies have been published in the last decade that correlate changes in Wnt ligands, receptors, gatekeepers, and downstream effector molecules to almost every type of human tumor. Dysregulation of both canonical and noncanonical Wnt signaling has been shown in numerous cancers [16,22,37,54,58,59,60,61,62,63]. Table 1 lists reports of Wnt pathway alterations in numerous tumor types.

## 4. Endometrial Cancer and Wnt Signaling

### 4.1. General Characteristics of Endometrial Cancer

Endometrial cancer is the most common gynecologic malignancy found in women worldwide and it is the second leading cause of gynecologic cancer death in the United States [87]; for 2021, the American Cancer Society predicts 66,570 new cases and 12,940 deaths [5]. Endometrial cancer rates have been steadily increasing in recent years. Although Caucasian women are slightly more likely to develop this disease than African American women, African American women are more likely to die from it [88]. Endometrial carcinoma mainly affects postmenopausal women aged 56 to 66 [89,90].

Increasing age, long-term use of unopposed hormone therapy, a history of nulliparity or infertility, irregular menstrual cycles, polycystic ovarian syndrome, obesity, diabetes mellitus, and hypertension are all risk factors for endometrial carcinoma [91,92,93,94,95,96,97,98]. In the United States, the five-year survival rate (at all stages) is estimated to be greater than 80% [6]. Patients with endometrial cancer have irregular endometrial bleeding, which frequently results in diagnosis at an early stage/grade in most cases. However, there are still many obstacles to overcome in the clinical treatment of endometrial cancer, particularly for patients diagnosed at a late stage or with a more aggressive histologic subtype.

The majority of endometrial carcinomas are produced from endometrial glands (adenocarcinomas), while the remainder are derived from the supporting stroma (sarcomas) (Figure 3). Endometrial carcinomas are classically grouped into classes based on their cause, clinical characteristics (histologic subtype), and pathogenesis [99,100]. Endometrioid endometrial carcinomas have a hyper-estrogenic condition, are typically diagnosed at a low grade, are minimally intrusive into the underlying uterine wall (myometrium) and have a better prognosis. They have estrogen and progesterone receptors and are usually receptive to progesterone therapy. These carcinomas most commonly occur before and around the time of menopause and account for roughly 80% of endometrial cancers. In the United States, endometrioid endometrial cancers mainly occur in white women, mostly in those with a background of endometrial hyperplasia. Fortunately, with surgical, hormone, and/or chemotherapy treatments, endometrioid endometrial carcinomas have an approximately 85% overall 5-year survival rate [100,101].

The second most common type of endometrial cancer is the serous subtype. These tumors are generally diagnosed at a high grade, are poorly differentiated with deep invasion into the underlying uterine wall (myometrium), are very aggressive, and have poor outcomes. They generally do not express the receptors for estrogen or progesterone and are not responsive to hormonal treatment. In the United States, serous endometrial cancers are more common in African American women. The 5-year survival rate for serous cancers is 53% [89,102]. Another less common subtype of carcinoma of the endometrial lining is clear cell carcinoma, representing about 6% of all endometrial cancers and with a 5-year survival rate of ~62% [100,101,103,104].

Understanding of the molecular profile of tumors may be more informative in terms of guiding treatment for individual endometrial tumors than segregation into histologic subtypes. The most comprehensive molecular study of endometrial cancer to date has been The Cancer Genome Atlas (TCGA) project, which revealed key molecular pathways and mutations associated with endometrial cancers; endometrioid subtypes most frequently had mutations of PTEN (a protein and lipid phosphatase), *CTNNB1* (β-catenin), PIK3CA, and KRAS, along with microsatellite instability and POLE mutations (Table 2) whereas serous endometrial carcinomas are characterized by genetic alterations in the tumor suppressor *TP53, HER2/neu, p16,* and E-cadherin [105,106,107,108].

### 4.2. Wnt Signaling Abbereations in Endometrial Cancer

Abnormal Wnt/β-catenin signaling plays an important role in endometrial cancer onset. Around 40% of endometrial cancers (mostly endometrioid histology) show abnormalities in the Wnt signaling pathway. Activation of the Wnt signaling pathway via alteration of the *APC*, *Axin*, *CTNBB1*, Wnt ligands, Wnt inhibitors Dickkopf3 (DKK3) and Dickkopf1 (DKK1), or the secreted Frizzled-related proteins may be a key driving event in the development of endometrial cancer. Nuclear accumulation of β-catenin (e.g., 13–69% of cases of endometrioid endometrial carcinoma) is the predominant alteration found in endometrioid tumors, but only a few studies have examined the role of Wnt signaling in its etiology [27,109,110].

Some studies showed that during the proliferative phase of the menstrual cycle β-catenin is mainly localized in the nucleus whereas during the secretory phase β-catenin translocated from the nucleus to the cytoplasm and cell membrane [28]. Estradiol induces Wnt4, Wnt5A, and Wnt7A ligands as well as FZD-2 receptor expression, subsequent stabilization of β-catenin in the cytoplasm, and the presence of active β-catenin in the nucleus [111,112]. However, progesterone can act as a profound inhibitor of Wnt/β-catenin signaling [19,113]. This may be due to the induction of DKK1 and FOXO1, which are important Wnt inhibitors. Studies by Wang et al. also showed that progesterone efficiently inhibits Wnt signaling in Ishikawa endometrial cancer cell lines [19].

Information on the importance of Wnt ligands in endometrial adenocarcinoma is increasing and it is likely that Wnt alteration plays a role in carcinogenesis. Bui et al. documented that Wnt4 mRNA is higher in normal endometrium than endometrial carcinoma; WNT2, WNT3, and WNT5A mRNA levels are similarly higher in normal endometrium than in endometrial carcinoma. This study suggests that down-regulation of WNT4, WNT2, WNT3, and WNT5A might be important in the development of endometrial cancer [102,114]. However, in 2014 Liu et al. reported that Wnt pathway components, such as Frizzled-10, TCF7, and LEF1 along with WNT5A, were overexpressed in βcatenin-mutated tumors in a validation study that compared the TCGA dataset to a broad independent cohort [115] (Figure 4).

Much of the research to date has focused on *WNT7* gene expression. The expression of *WNT7A* and *WNT7B* genes is high in endometrial carcinoma cell lines but not in normal primary endometrial cultures [32,116,117]. In vivo, expression of both *WNT7A* and *WNT7B* can be detected in normal human endometrial tissues and human endometrial tumors. This discrepancy is likely due to the artificial environment and regulatory signals that control *WNT7A* and *WNT7B* expression [114]. In one clinical study of endometrial carcinoma, the expression of *WNT7A* gene was absent or reduced and was negatively linked with FIGO stage, grade, lymph node metastasis, depth of myometrial invasion, and peritoneal cytology in approximately 60% of patients [116]. In a large-scale population study of 244 endometrioid endometrial cancer patients, the *WNT7A* gene was overexpressed in most cases of endometrial cancer [117]. It is important to note, however, that the lack of expression of *WNT7A* positively correlated with overall survival and disease-free survival in endometrial cancer patients [117,118], suggesting that overexpression of *WNT7A* plays a pathogenic role.

In addition to *WNT7A*, the expression of *WNT10A* and *WNT10B* ligands has been associated with estrogen-related carcinogenesis of endometrial cancer. Wnt10b protein expression is significantly higher in endometrial cancer tissue in comparison to hyperplastic endometrium and normal endometrium [119]. Moreover, the expression of Wnt10b correlates with the histological type of cancer, cell maturity, FIGO stage, and lymphatic metastasis. Elevated Wnt10b protein expression is associated with a better prognosis in endometrial patients. Studies in vitro have also proved that Wnt10b enhances proliferation and suppresses apoptosis by activation of β-catenin and c-Myc as well as APC inhibition [102,119]. However, the significance of the *WNT10A* gene in endometrial cancer, aside from its upregulated expression, remains unknown [120].

Studies by van der Zee et al. demonstrated a synergistic effect of the Wnt/β-catenin and PTEN pathways in endometrial cancer [121]. Loss of *PTEN* function is linked with endometrial cancer onset and *PTEN* is frequently mutated in endometrioid endometrial tumors [9]. *PTEN* function loss stimulates defects in the function of the *APC* that normally regulate the formation of the β-catenin “destruction complex” that in turn causes constitutive activation of the Wnt/β-catenin pathway and promotes the development of cancer. The result of the constitutively activated Wnt/β-catenin pathway is squamous cell metaplasia (SCM) with no malignant transformation. Consequently, it appears that activation of the Wnt/β-catenin pathway accelerates rather than initiates the cancer while the simultaneous loss of PTEN activity and activation of the Wnt/β-catenin pathway is associated with the development of an aggressive form of endometrial cancer [121].

The other major alteration reported in endometrial cancers is nuclear accumulation of β-catenin (*CTNNB1*). Between 10% and 45% of endometrial cancers have missense mutations of *CTNNB1* [70,73,122]. *CTNNB1* mutations are apparently found in the early stages of endometrial carcinogenesis [70]. Many studies show that *CTNNB1* mutation is primarily detected in endometrioid endometrial cancer rather than nonendometrioid endometrial carcinoma cases (NEEC) [26,69,122,123].

β-catenin mutations in exon 3, on the serine/threonine residue that is the site of regulatory phosphorylation by GSK3β, have been identified [24,124]. Mutations within this domain prevent β-catenin degradation and result in β-catenin nuclear accumulation in endometrial tumors. Of 76 uterine endometrial carcinoma cases examined, 10 cases (approximately 13%) had mutations within this domain; 20 cases (approximately 26%) without β-catenin mutations showed similar accumulation of the protein [24]. In total, 38% of endometrial carcinoma cases showed nuclear accumulation. Those cases that showed β-catenin accumulation but not the exon 3 β-catenin mutation may have carried alterations in β-catenin outside of exon 3, alterations to APC, or had the involvement of members of the Wnt family proteins that elevate β-catenin expression [125,126,127]. This study showed that nuclear accumulation of beta-catenin, due to β-catenin mutations in exon 3 or alternative mechanisms, may play a significant role in development of endometrial carcinomas. Travaglino et al. and Nei et al. observed a 10% mutation frequency for β-catenin in endometrial cancer (2/20 tumors) while showing that 30% of endometrial cancer specimens exhibited nuclear β-catenin accumulation. They found very intense nuclear staining of β-catenin more often in endometrial hyperplasia than in endometrial carcinoma samples [28,29]. Ikeda et al. and others reported an 11% somatic mutation frequency (5/44 tumors) and that the tumors with mutations exhibited accumulation of the β-catenin in the cytoplasm and nucleus [24,28,122]. Mirabelli-Primdahl et al. identified a 45% *CTNNB1* mutation frequency in 29 endometrial cancer tumors with or without microsatellite instability (33% MSI-H and 50% MSS/MSI-L), suggesting that there is no correlation with the presence or absence of underlying microsatellite instability [128]. Schlosshauer et al. and Saegusa have reported slightly higher β-catenin mutation frequencies (18% and 23%, respectively) in endometrioid endometrial cancers [27,91], while Moreno-Bueno observed 11% *CTNNB1* mutation frequency in endometrioid endometrial cancer [129]. Recently, Antonio et al. showed that exon 3 *CTNNB1* mutations and concomitant nuclear expression of β-catenin were found in 16.8% of the 125 endometrial carcinomas studied. Nuclear localization of β-catenin in neoplastic cells varies between 5% and 60% in *CTNNB1* mutant endometrial carcinomas (mean 19.8%) [130]. Another study explored the impact of ARID1A and CTNNB1/-catenin alterations in a group of molecularly classified endometrium cancers using targeted next-generation sequencing (NGS) [131]. Kim et al. showed that exon 3 *CTNNB1* mutations were found in 63 (18%) of 345 endometrial cancer patients; 53 of these patients had tissue available for immunohistochemistry. Among these 53 samples, 46 were of pure endometrioid histology and seven were mixed endometrioid and nonendometrioid carcinomas. This study found that 45/53 (85%) of *CTNNB1* mutant endometrial cancers had nuclear β-catenin localization [132]. Taken together, these findings suggest that total nuclear β-catenin may not be as important as previously thought in driving the poor prognosis seen in *CTNNB1*-mutant endometrial cancers as other cellular components may also help activate the Wnt pathway. Detailed *CTNNB1* mutational rates are listed in Table 3.

Another study by Shelton and Goodheart et al. examined the role of lymphoid enhancer-binding factor 1 (Lef1) in endometrial cancer [133]. Lef1, a member of the T cell factor (TCF)/Lef1 family of high-mobility group transcription factors, is a downstream mediator of the Wnt/β-catenin signaling pathway, although it can also modulate gene transcription independently. Overexpression of Lef1 has been detected in several cancers, principally colon and colorectal cancers, leukemia, melanoma, and pancreatic cancer, making it a valuable biomarker in predicting patient prognosis. Lef1 is overexpressed in endometrioid endometrial carcinomas compared to nonendometrioid endometrial carcinomas, and where the Lef1 downstream targets are activated through cyclin D1 and MMP7 [134]. An ongoing clinical study (NCT03787056, www.clinicaltrials.gov (accessed on 10 May 2021)), published in Dec 2018 and sponsored by Hospices Civils de Lyon [135], is investigating the predictive value of progastrin titer at diagnosis and of progastrin kinetics during treatment in cancer patients (approximately 410 participants). Progastrin is a prohormone that under physiological conditions is transformed into gastrin in the G cells of the stomach. Gastrin is a hormone that excites the production of gastric acids during digestion. It is also vital for the regulation of gastric mucosal cell growth. Progastrin is not present in the peripheral blood of a healthy person. However, patients with endometrial cancer and other cancers (colorectal, gastric, ovarian, breast, cervix uterus, and melanoma among others) have abnormally high progastrin blood levels. The WNT/ß-catenin oncogenic pathway has a direct target gene, GAST, which codes for progastrin [135]. Progastrin, as measured in the peripheral blood of cancer patients undergoing treatment, may thus be a new effective marker for cancer diagnosis and prognosis at various stages.

## 5. Pharmacological Wnt Inhibitors and Clinical Trials

Precision medicine is often limited by the identification of biological mediators, the pathology of various conditions, and the development of treatments that efficiently target specific genes; in addition, treatments often have side effects. Recent research has significantly increased our understanding of the function of Wnt secretion in carcinogenesis and revealed novel therapeutic targets. The Wnt cascade constitutes potential targets for pharmacological intervention and use in personalized medicine; however, the role of the individual β-catenin pathway components must be better elucidated in endometrial cancer. Currently, very few effective treatment strategies exist for advanced endometrial cancer patients who have failed traditional chemotherapy. Novel biologics targeting VEGF and mTOR pathways have shown promising results for endometrial cancer in phase II clinical trials [136,137]. CTNNB1 mutations and epigenetic silencing of negative Wnt regulators contributes to the aberrant activation of β-catenin and this process can be attenuated at different cellular levels, hence there is a clear need for drugs that interfere with the transcriptional functions of β-catenin [138,139]. Emerging evidence indicates that β-catenin-dependent signaling plays a crucial role in endometrioid endometrial cancer progression and nuclear β-catenin could be a novel therapeutic target of endometrial cancer treatment.

Currently, numerous β-catenin-dependent signaling inhibitors are available as pre-clinical, trial, or FDA-approved therapies, including MPA, DKN-1, OMP-54F28, Niclosamide, and PRI-724 among others.

### 5.1. Medroxyprogesterone Acetate (MPA)

Wang et al., in a study of 21 patients, showed that a synthetic form of progesterone (medroxyprogesterone acetate, MPA) counteracts the proliferative effects of E2 (estradiol) during the normal menstrual cycle, hyperplasia, and early endometrial carcinogenesis by inhibiting Wnt/β-catenin signaling gene expression [19]. While lesions are expected to disappear with extended use of MPA treatment, when therapy is stopped there is marked recurrence as documented by the Yahata research group in a limited study of six young patients [140].

### 5.2. Levonorgestrel Intrauterine Device

In a clinical study, Westin and his research team (clinicaltrials.gov, NCT00788671) documented the effectiveness of a levonorgestrel-releasing intrauterine system in treating patients with complex atypical hyperplasia or grade I endometrial cancer [141]. The aim was to explore hormone replacement therapy using levonorgestrel, a type of progesterone, to combat endometrial cancer. A total of 57 patients were treated (21 endometrial cancer, 36 complex atypical hyperplasia); median age was 48 years and the median BMI was 45.5 kg/m^2^. Of 47 evaluable patients, the 12-month response rate was 83%, 37 were complete responders (8 endometrial cancer, 29 complex atypical hyperplasia), 2 were partial responders (2 endometrial cancer), 3 had stable diseases (2 endometrial cancer, 1 complex atypical hyperplasia), and 5 had progressive diseases (3 endometrial cancer, 2 complex atypical hyperplasia). Westin has also reported on genes that are important in cell proliferation, estrogen signaling, and Wnt signaling [142].

### 5.3. DKN-01

DKN-01 is a humanized monoclonal antibody (Mab) targeting Dickkopf-1 (DKK1) and is being developed as an antineoplastic agent. One study (NCT03395080), started in Feb 2018 and funded by Leap Therapeutics, Inc., uses DKN-01 as a monotherapy or in combination with paclitaxel in patients with recurrent epithelial endometrial or epithelial ovarian cancer or carcinosarcoma [143]. Doses of 300mg DKN-01 will be infused in a total of 124 patients in combination with paclitaxel (18 years and older). In this study, they will also observe response to therapy in patients with and without activating β-catenin mutations and/or Wnt signaling genetic alterations and with recurrent EEC (epithelial endometrial cancer) or EOC (epithelial ovarian cancer) or carcinosarcoma.

### 5.4. Porcupine Inhibitor

Wnt-driven cancers can be targeted at many points in the pathway [23,57,138]. One strategy is to prevent the secretion of all Wnts by inhibiting the activity of Porcupine (PORCN), an endoplasmic reticulum-resident enzyme that palmitoleates Wnts at a highly conserved serine residue post-translationally [144]. Both Wnt secretion and binding to Frizzled receptors are dependent on palmitoleation [145]. Inhibition of PORCN enzymatic activity provides a way to overcome the limitations of β-catenin inhibitors that can only hinder the canonical Wnt signaling cascade or anti-Frizzled antibodies, which cannot attack all Frizzled receptors [144,146]. Blocking a significant post-translational modification, palmitoleation, provides a useful therapeutic intervention [144]. The Madan research group developed a novel, potent, orally available PORCN inhibitor, ETC-1922159 (henceforth called ETC-159) that blocks the secretion and function of all Wnts. RSPO translocations are seen in ~4–18% of patients with ovarian, endometrial, and gastric cancer, suggesting that ETC-159 may be a beneficial therapeutic strategy for EC treatment [144]. A clinical trial (Phase 1A/B) is ongoing, employing 83 participants (NCT02521844) and sponsored by the EDDC (Experimental Drug Development Centre), to evaluate the safety and tolerability of ETC-159 in different advanced solid tumors [147]. LGK974, another PORCN inhibitor that has been shown to suppress tumor progression, is currently being evaluated in clinical trials for solid malignancies, including esophageal squamous-cell carcinoma, pancreatic adenocarcinoma, and BRAF-mutated colorectal cancer [148].

### 5.5. OMP-54F28

OMP-54F28 is a fusion protein consisting of the extracellular ligand-binding domain of Fzd8 and a human immunoglobulin G1 (IgG1) Fc domain. This decoy receptor interferes with Wnt signaling by sequestering secreted Wnts and shows antitumor activity in several patient-derived xenograft (PDX) models [149,150]. Currently, three phase 1b trials of OMP-54F28 in combination with chemotherapeutics in liver, ovarian, and pancreatic cancer are ongoing. Furthermore, a phase I clinical trial testing the safety of OMP131R10, a RSPO3-binding antibody, in advanced solid tumors and metastasized colorectal cancer has recently been completed, although results are as yet unpublished [151]. Several clinical phase I studies have documented Ipafricept (OMP-54F28) as a potent agent to target Wnt ligands in patients with advanced solid tumors [152,153].

### 5.6. Niclosamide

Niclosamide (trade name Niclocide), an FDA-approved salicylamide derivative used for the treatment of tapeworm infections, targets the Wnt/β-catenin pathway. Recently, it was observed that niclosamide inhibits the β-catenin-dependent pathway in ovarian tumor cells isolated from 34 patient ascites [154]. Moreover, the inhibitory effects of niclosamide on Wnt/β-catenin signaling were also found in primary human glioblastoma cells [155]. Niclosamide promoted LRP5/6 degradation [156], Wnt receptor Fzd1 endocytosis [157], downregulation of Dvl2 proteins, and inhibition of Wnt3A-stimulated β-catenin stabilization and TCF/LEF reporter activity [157,158] as well as enhanced cytotoxicity in all the patient derived tumor spheres. Niclosamide has also been shown to interfere with Wnt7/β-catenin signaling and can reduce tumor growth in a xenograft mouse model [159]. It may be worthwhile to start an in vitro study or a clinical trial in endometrial cancer patients with niclosamide therapy.

### 5.7. PRI-724 and ICG-001

Besides the intracellular perturbation of Wnt secretion and ligands, an inhibitor of the downstream Wnt pathway is currently undergoing clinical trials. Emami et al. developed a small molecule inhibitor (PRI-724) closely related compound ICG-001 that targets the complex formation of β-catenin and CBP that specifically downregulates the expression of β-catenin-TCF-responsive genes [160]. Treatment with PRI-724 therefore interferes with Wnt/β-catenin signaling and inhibits tumor growth [161]. PRI-724 has been shown to have an acceptable safety profile in a phase 1 clinical trial and is now under further clinical investigation [162]. A phase 2 trial of PRI-724 in combination with bevacizumab is now being planned for metastatic colorectal carcinoma patients (clinicaltrials.gov, 2015, NCT02413853). It may be worthwhile to conduct an in vitro study or a clinical trial in endometrial cancer patients with this PRI-724 and ICG-001 treatment therapy [163].

### 5.8. Salinomycin

Salinomycin, an antibiotic potassium ionophore, is a selective inhibitor of breast cancer stem cells [164]. Salinomycin induces apoptosis, interferes with Wnt/β-catenin signaling, and consequently inhibits proliferation, migration, and invasion as well as shows antitumor effects on side population (SP) cells obtained from invasive endometrial cancer cells [165]. Therefore, it is important to examine either salinomycin alone or in combination with other drugs as it could significantly interfere with proliferation, apoptosis, migration, and invasion in human endometrial cancer cells as well as in endometrial cancer stem cells.

### 5.9. Curcumin

Curcumin inhibits the initiation, promotion, and progression of carcinogenesis. Curcumin may exert anticancer effects through a variety of pathways that are involved in mutagenesis, apoptosis, tumorigenesis, cell cycle regulation, and metastasis. The anticancer effects of curcumin involve in the activation of apoptotic pathways as well as in inhibition of tumor microenvironments (inflammation, angiogenesis, and tumor metastasis). Extensive studies have demonstrated that curcumin targets several therapeutically relevant cancer signaling pathways, such as p53, Ras, PI3K, AKT, Wnt/β-catenin, and mTOR [166]. Clinical trials also suggest that curcumin either alone or as part of a combination with other drugs possesses promising anticancer effects in cancer patients without toxic side effects [167,168]. Feng et al. documented that Curcumin promotes apoptosis in human endometrial carcinoma cells (RL-952) by downregulating the expression of androgen receptors (ER) through inhibition of Wnt signaling [169].

### 5.10. miRNA Treatment

MicroRNAs (miRNAs) are small noncoding RNAs comprised of about 22 nucleotides that are a class of naturally occurring small noncoding RNA molecules that regulate cellular function by controlling multiple target messenger RNAs. A single miRNA may target at least 200 genes and a single gene can be regulated by many RNAs. Since individual miRNAs have potentially numerous target genes, miRNA dysregulation can profoundly regulate the cellular machinery and facilitate cancer hallmarks [170]. Numerous pieces of research have proved that dysregulation of microRNAs enhance tumorigenesis and metastasis. Furthermore, recent studies have shown that miRNAs can act as either potent oncogenes or tumor-suppressor genes. To understand the role of miRNAs in endometrial cancer, all miRNAs where regulation has been shown are extensively reviewed in this section.

Wang et al. documented that miR-15a-5p levels are significantly decreased in endometrial cancer compared with healthy controls both in in vitro and in vivo [171]. In this study, endometrial cancer tissues were obtained from patients who underwent a hysterectomy. The tissue samples included eight endometrial cancer tissues and three normal control tissues. They confirmed that miR-15a-5p suppresses endometrial cancer cell growth via Wnt/β-catenin signaling by inhibiting WNT3A [171]. Overexpression of miR-21 has also been reported in endometrioid endometrial cancer. Xiaoyan et al. demonstrated that miR-21 expression was inversely correlated with PTEN protein expression in the KLE cell line. The overexpression of miR-21 results in downregulation of PTEN, leading to the modulation of endometrioid endometrial cancer cell proliferation [172]. Over the last decade, several researchers have documented that miR-21 promotes progression of numerous cancers via the Wnt/β-catenin pathway [173,174,175]. Additionally, miR-146b-5p is dramatically decreased in the endometrial carcinoma cell line Ishikawa compared with normal endometrial cells. Progesterone has been shown to regulate the NEAT1/miR-146b-5p axis via the Wnt/β-catenin signaling pathway. In this study, progesterone exerted suppressive influence on endometrial cancer progression via regulation of the lncRNA NEAT1/miR-146b-5p-mediated Wnt/β-catenin signaling pathway, potentially revealing new strategies for developing more effective therapeutics [176]. Although, the miR-200 family (a/b/c) has been shown to be downregulated in most type of cancers, the upregulation seen in endometrial carcinoma is also seen in ovarian carcinoma, melanoma, and colorectal carcinoma [177,178]. In addition, miR-200a has been shown to directly interact with the 3′ UTR of *CTNNB1* to suppress Wnt/β-catenin signaling in SGC790 and U251 cells [179]. Moreover, downregulation of miR-200a promotes epithelial–mesenchymal transition in gastric adenocarcinoma and brain meningiomas and arachnoidal cells [180,181]. Thus, the function of upregulated microRNA-200a and its relationship to Wnt signaling in endometrial cancer is still unknown.

To date, the use of various inhibitors of Wnt/β-catenin targeting cancer as prognostic markers and tumor suppressors has been reported and novel agents targeting the Wnt/β-catenin signaling pathway have shown encouraging results. Endometrial cancer should be the focus of these initiatives with the goal of establishing Wnt pathway components as prognostic and predictive biomarkers as well as demonstrating preclinical evidence and assessing the efficacy of established inhibitors for this pathway.

## 6. Wnt Signaling Crosstalk: An emerging Area for Drug Development

The Wnt signaling pathway has been shown to be involved in crosstalk with several other signaling pathways, such as the Notch, mTOR, Hedgehog, and estrogen signaling pathways, and with the tissue microenvironment. Since endometrial cancer pathogenesis involves various genomic changes and aberrant signaling pathways; investigating the association between the incidence of these molecular disorders and their plausible crosstalk will potentially reveal important targets for synergistic drug combinations in endometrial cancer and advance the development of targeted and more successful therapies [182,183].

Cancer stem cell (CSC) activation of the Wnt pathway has been shown to initiate a complex transcriptional program resulting in the expression of genes involved in drug resistance (such as ABCB1 and CD44) and immune function (such as PD-L1 and CD47) [184]. A recent study identified SPARC-related modular calcium-binding 2 (SMOC–2) as a novel endometrial cancer stem cell signature gene and demonstrated that SMOC-2 is involved in activation of the Wnt/β-catenin pathway [185]. Furthermore, salinomycin, an antibacterial and coccidiostatic therapeutic drug that is a selective inhibitor of CSCs, has been shown to downregulate the expression of Wnt target genes such as LEF1, cyclin D1, and fibronectin [165]. Given the prominent role of the Wnt signaling cascade in stem cell biology, the study of endometrial cancer stem cells seems to be a promising area of central focus in the near future and has prospects for safely and effectively targeting this signaling to abolish the cancer stem cell population in endometrial cancer.

Wnt signaling acts as a bridge between tumor cells and the tumor microenvironment contributing to cancer progression, drug resistance, and immunity escape [186]. While Wnt signaling plays a major role in cellular homeostasis, regulating immune cell development and function, Wnt ligands secreted by stromal cells and inflammatory cells present in the tumor microenvironment promote tumor invasion, metastasis, and tolerance [186]. Therefore, the role of Wnt signaling in immune cells is becoming an area of active research. With the recent progress in immunotherapy, the combination therapy with modulation of Wnt/β-catenin signaling is expected to overcome primary, adaptive, and acquired resistance to cancer immunotherapy and to add therapeutic benefit in personalized molecular therapy for women with advanced and recurrent endometrial cancer [187,188]. A recent study demonstrated tumor necrosis factor receptor-associated factor 6 (TRAF6) as a key regulatory factor for Wnt3a-induced regulation of β-catenin and subsequent activation of Wnt3a target genes implicated in prostate cancer progression, suggesting TRAF6 as an important novel target for inhibiting cancer [189].

Increasing evidence for the crosstalk between Wnt/β-catenin and autophagy suggests the interplay between both pathways as an attractive therapeutic target [190]. Hydroxychloroquine, chloroquine, bafilomycin, and concanamycin, which are well known autophagy inhibitors, have been shown to inhibit Wnt/β-catenin signaling via inhibiting v-ATPase in CRC, suggesting their use as cancer-specific Wnt/β-catenin inhibitors [148,191,192]. However, further research is warranted regarding the safety and efficacy in endometrial cancer.

Unopposed estrogen and Wnt signaling aberrations are major pathways accounting for the endometrial cancers. A study using a uterine epithelial cell specific inducible Cre mouse model and 3D in vitro culture of human endometrial cancer cell lines demonstrated interplay between hormone and Wnt signaling pathways in endometrial cancer. This study showed that activating mutations in the Wnt signaling pathway for a prolonged period is insufficient to cause endometrial cancer but leads to development of endometrial hyperplasia. However, together with unopposed estrogen, activating mutations in Wnt signaling drive the progression of endometrial hyperplasia to endometrial cancer. Overall, this study indicated use of progesterone as a targeted therapy for endometrial cancer patients with an activated Wnt signaling pathway [193]. Another study demonstrated progesterone-mediated regulation of the Wnt/β-catenin signaling pathway via regulation of lncRNA NEAT1/miR-146b-5p contributing to endometrial cancer growth inhibition [176]. Feng et al. also showed that progesterone treatment suppressed the Wnt signaling as well as β-catenin via suppressing the level of H19, which directly targets miR-152 [194]. Altogether, these studies unravel new strategies for developing more effective therapeutics. However, there is still a lack of studies evaluating the combined effect of progesterone and Wnt signaling inhibitors.

## 7. Conclusions and Future Perspective

Abnormal activation of WNT signaling has been reported in the majority of type-1 endometrial cancer cases with β-catenin mutations in 20–25% of cases. Given the current lack of treatment options for advanced and recurrent endometrial cancer patients and the growing body of evidence supporting the role of Wnt signaling at early stages of endometrial carcinogenesis, Wnt signaling represents a promising intervention for targeted therapies in endometrial cancer patients. Various inhibitors targeting different molecules of this pathway have been developed, though only a few studies have addressed the effects of Wnt inhibitors in endometrial cancer and they are still at an early phase and far away from clinical trials. Future studies evaluating the efficacy and safety of Wnt/β-catenin signaling inhibitors in endometrial cancer are needed to aid in the development of targeted novel therapies that can reverse the increasing trend of mortality in endometrial cancer patients. With the recent advances in genome editing, further research utilizing human-derived organoid models and tissue-specific inhibition of Wnt signaling will enhance our understanding and identify novel targets of the Wnt/β-catenin signaling pathway in endometrial cancer. Since combination therapy with other cancer drugs has been shown to limit off-target toxicities and considering the contribution of the crosstalk between Wnt signaling and the host immune response to therapeutic resistance and immune escape, Wnt inhibitors in combination with immunotherapies must be evaluated in endometrial cancer to improve both overall treatment efficacy and patient outcomes.

## Figures and Tables

**Figure 1 cancers-13-02351-f001:**
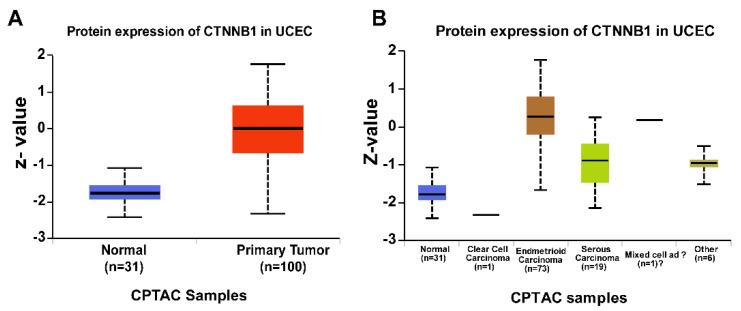
*CTNNB1* proteomic expression profile in normal endometrium tissue vs. uterine corpus endometrial carcinoma (UCES, (**A**)), and across different histological grades (**B**).

**Figure 2 cancers-13-02351-f002:**
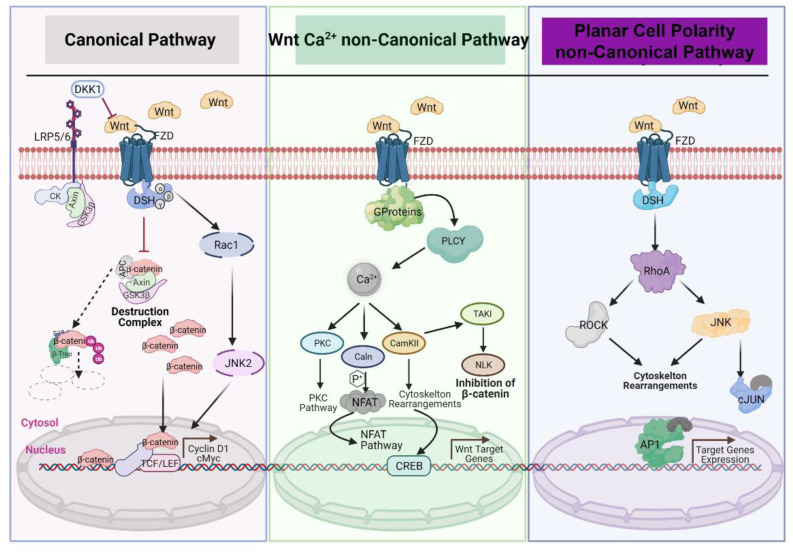
Schematic representation of Wnt signaling.

**Figure 3 cancers-13-02351-f003:**
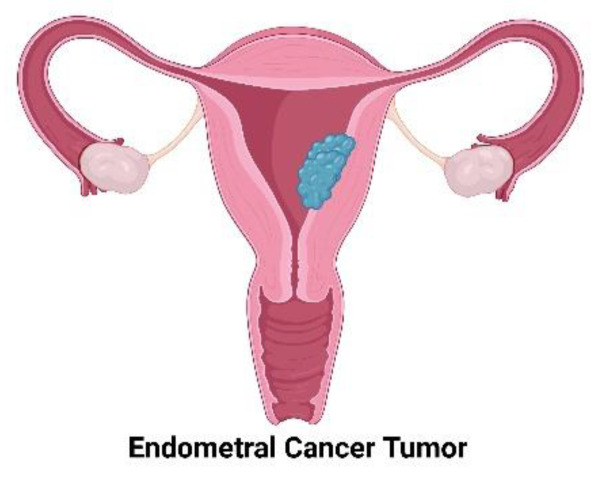
Endometrial cancer begins in the layer of cells that form the inner lining of the uterus (endometrium).

**Figure 4 cancers-13-02351-f004:**
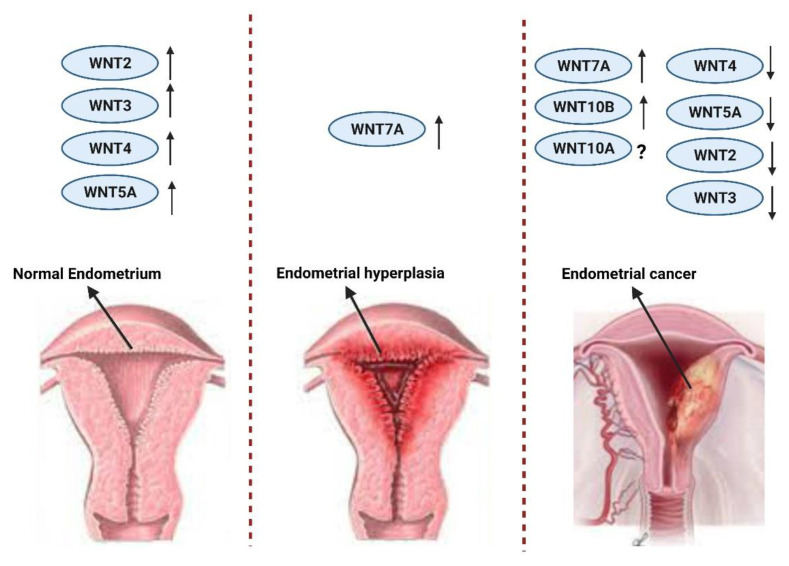
Expression of Wnt ligands in normal endometrium, hyperplasia, and endometrial cancer.

**Table 1 cancers-13-02351-t001:** Alterations of major component of Wnt signaling pathway in Various Tumor Types.

S. No	Type of Cancer	Mutation(s)	% Mutated	Reference(s)
1	Colorectal adenocarcinoma	APC	60%	[64,65,66,67,68]
		TCF7L2	7.70%	
		AMER1	10%	
		AXIN1	11%	
		CTNNB1	6.50%	
2	Endometrial carcinoma	CTNNB1	19–30%	[26,69,70]
3	Cervical carcinoma	WIF1	60%	[71]
4	Ovarian carcinoma	CTNNB1	6.4	[54,72,73]
		APC	29%	
5	Prostatic	CTNNB1	5%	[74,75,76]
	adeno	TMPRSS2-ERG gene	55%	
	carcinoma	fusion		
6	Wilms tumor	WTX	32%	[77]
		CTNNB1	15%	
		WT1	12%	
7	Breast carcinoma	APC	2.2	[58,60]
		Aberrant mRNA		
		splicing of LRP5		
8	Hepatocellular	CTNNB1	9.60%	[78,79,80,81]
	carcinoma	AXIN1	8.6	
		APC	80%	
9	Pancreas ductal Carcinoma	APC	4.8	[82,83,84,85,86]
		CTNNB1	1.6	
		RNF43	1.5	

**Table 2 cancers-13-02351-t002:** Molecular features of endometrial carcinoma: endometrioid and serous endometrial carcinomas.

Molecular Markers	Endometrioid Endometrial Cancer	Serous Endometrial Cancer
Microsatellite instability	30%	0–5%
DNA mismatch p16	10%	45%
p53 mutation	10%	90%
KRAS mutation	20–40%	0–5%
ER/PR expression	70–80%	5%
Her-2 amplification/overexpression	15–20%	18–45%
PTEN mutation	40–50%	10%
β-catenin mutation	14–44%	0–5%
*CCNE1* amplification	0–5%	30%
PIK3CA mutation	40%	15%

**Table 3 cancers-13-02351-t003:** β-Catenin mutations in endometrial cancer.

Study	Mutation Frequency of Beta Catenin in Endometrial Cancer (%)	Ref.
Fukuchi et al. (1998)	13	[24]
Nei et al. (1999)	10	[28]
Mirabelli-Primdahl et al. (1999)	45	[128]
Ikeda et al. (2000)	11	[122]
Moreno-Bueno et al. (2002)	11	[129]
Machin et al. (2002)	21	[26]
Schlosshauer et al. (2000)	19	[27]
Saegusa et al. (2001)	23	[91]
Antonio et al. (2021)	16.8	[130]
Kim et al. (2018)	18	[132]
Kurnit et al. (2017)	18	[70]

## Data Availability

Not Applicable.

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
