# Peer review of "Targeting Wnt Signaling in Endometrial Cancer"

_cancers, 2021, doi:10.3390/cancers13102351_

Round 1

Reviewer 1 Report

This review is interesting because it summarizes the mporntace of the WNT pathway in endometrial carcinoma. 
I suggest revision of minor typos (there are several in the text).
I strongly suggest a table with CTNNB1 mutation rates in endometrial carcinoma in recent and not dated work as reported by the authors.
I suggest an immunohistochemical histopathology image showing nuclear translocation of B-Catenin in endometrial carcinoma.
I certainly strongly suggest citing some important work also recently published in Cancers that evaluated CTNNB1 in endometrial carcinoma:

  • De Leo, Antonio et al. “ARID1A and CTNNB1/β-Catenin Molecular Status Affects the Clinicopathologic Features and Prognosis of Endometrial Carcinoma: Implications for an Improved Surrogate Molecular Classification.” Cancers vol. 13,5 950. 25 Feb. 2021, doi:10.3390/cancers13050950
  • Joehlin-Price, Amy et al. “Molecularly Classified Uterine FIGO Grade 3 Endometrioid Carcinomas Show Distinctive Clinical Outcomes But Overlapping Morphologic Features.” The American journal of surgical pathology vol. 45,3 (2021): 421-429. doi:10.1097/PAS.0000000000001598

All gene abbreviations should be put in italics.
Figures should be of higher quality.

Author Response

This review is interesting because it summarizes the importance of the WNT pathway in endometrial carcinoma. 

We thank Reviewer 1 for his praise of our work. We agree with his minor comment and accordingly modified our manuscript.

I suggest revision of minor typos (there are several in the text).

Response: As suggested, we revised whole manuscript for typos.

I strongly suggest a table with CTNNB1 mutation rates in endometrial carcinoma in recent and not dated work as reported by the authors.

Response: As suggested we updated table 3: CTNNB1 mutation rates in endometrial carcinoma with recent publication.

I suggest an immunohistochemical histopathology image showing nuclear translocation of B-Catenin in endometrial carcinoma.

Response: We agreed with Reviewer’s comment that immunohistochemical histopathology image showing nuclear translocation of B-Catenin in endometrial carcinoma will add to this review however, due to copyright issue we are unable to add similar figure at this moment. However, we added reference for publication showing nuclear translocation of B-Catenin in endometrial carcinoma. Also, we added a figure (Figure 1, Page 3) showing increased expression of β-catenin in human endometrial cancer patient from Cancer Genome Atlas database.

I certainly strongly suggest citing some important work also recently published in Cancers that evaluated CTNNB1 in endometrial carcinoma:

  • De Leo, Antonio et al. “ARID1A and CTNNB1/β-Catenin Molecular Status Affects the Clinicopathologic Features and Prognosis of Endometrial Carcinoma: Implications for an Improved Surrogate Molecular Classification.” Cancers vol. 13,5 950. 25 Feb. 2021, doi:10.3390/cancers13050950
  • Joehlin-Price, Amy et al. “Molecularly Classified Uterine FIGO Grade 3 Endometrioid Carcinomas Show Distinctive Clinical Outcomes But Overlapping Morphologic Features.” The American journal of surgical pathology vol. 45,3 (2021): 421-429. doi:10.1097/PAS.0000000000001598

Response: As suggested, we added these and several other recent publications throughout the manuscripts.

All gene abbreviations should be put in italics.

Response: As suggested, we made all gene abbreviations in italics.

Figures should be of higher quality.

Response: As suggested, we replaced figure with high quality figures.

Reviewer 2 Report

The review article is interesting and sheds light on the role of wnt signaling in endometrial cancer. The authors could address the below comments to improve the article. 

  1. Figure 1 is very illustrative, It looks like wnt ca2 pathway is the only non canonical pathway in the figure, the authors could both Wnt Ca2 and PCP pathway as non canonical pathways.
  2. They need to include a future perspective on various molecular targets that are emerging in the field of wnt signaling such as cross talk with inflammatory ligases such as TRAF6 and others.
  3. With advancement of genome editing, it is important to include in the conclusion.

Author Response

The review article is interesting and sheds light on the role of wnt signaling in endometrial cancer. The authors could address the below comments to improve the article. 

We thank Reviewer 2 for his praise of our work. We agree with his comment and accordingly modified our manuscript.

  1. Figure 1 is very illustrative, It looks like wnt ca2 pathway is the only non canonical pathway in the figure, the authors could both Wnt Ca2 and PCP pathway as non canonical pathways.

Response: As suggested, we revised figure 1.

  1. They need to include a future perspective on various molecular targets that are emerging in the field of wnt signaling such as cross talk with inflammatory ligases such as TRAF6 and others.

Response: As suggested, we added a complete section as “Wnt Signaling crosstalk: An emerging area for drug development” and future perspective with conclusion.

  1. With advancement of genome editing, it is important to include in the conclusion.
    Response: As suggested, we updated conclusion.

Round 2

Reviewer 1 Report

The authors have consistently followed the advice of the reviewers, and the updated version of the manuscript represents an interesting and comprehensive review of the WNT pathway in endometrial cancer.